# ON A BENEFIT OF MASKED LANGUAGE MODEL PRE-TRAINING: ROBUSTNESS TO SIMPLICITY BIAS

## ABSTRACT

Despite the success of pretrained masked language models (MLM), why MLM pretraining is useful is still a question not fully answered. In this work we theoretically and empirically show that MLM pretraining makes models robust to lexicon-level spurious features, partly answering the question. Our explanation is that MLM pretraining may alleviate problems brought by simplicity bias (Shah et al., 2020), which refers to the phenomenon that a deep model tends to rely excessively on simple features. In NLP tasks, those simple features could be token-level features whose spurious association with the label can be learned easily. We show that MLM pretraining makes learning from the context easier. Thus, pretrained models are less likely to rely excessively on a single token. We also explore the theoretical explanations of MLM's efficacy in causal settings. Compared with Wei et al. (2021), we achieve similar results with milder assumptions. Finally, we close the gap between our theories and real-world practices by conducting experiments on real-world tasks.

## 1 INTRODUCTION

The question "*why is masked language model (MLM) pretraining (Devlin et al., 2019; Liu et al., 2019) useful?*" has not been totally answered. In this work, as an initial step toward the answer, we show and explain that MLM pretraining makes the model robust to lexicon-level features that are spuriously associated with the target label. It gives the model a better generalization capability under distribution shift.

Previous studies have empirically shown the robustness of MLM pretrained models. Hao et al. (2019) show that MLM pretraining leads to wider optima and better generalization capability. Hendrycks et al. (2020) and Tu et al. (2020) show that pretrained models are more robust to out-of-distribution data and spurious features. However, it remains unanswered why pretrained models are more robust.

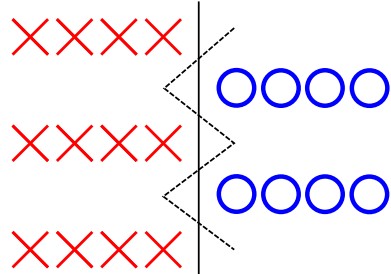

Figure 1: The pitfall of simplicity bias Shah et al. (2020): The solid line is a simple (linear) decision boundary that utilizes only one dimension, while the dashed line is a more complex decision boundary that utilizes two dimensions and maximizes the margin.

We conjecture that models trained from scratch suffer from the pitfall of simplicity bias Shah et al. (2020) (Figure 1). Shah et al. (2020) and Kalimeris et al. (2019) showed that deep networks tend to converge to a simple decision boundary that involves only a few features. The networks may not utilize all the features and thus may not maximize the margin, which results in worse robustness. A consequence of this could be that a model may excessively rely on a feature that has spurious association with the label and ignore the other features that are more robust. In the studies of Shah et al. (2020) and Kalimeris et al. (2019), they investigated networks with continuous input. Lovering et al. (2021) discovered similar results on synthetic NLP tasks, where the inputs are discrete. We will further explore this discrete setting in this work.

We start the exploration with the following assumptions: Let the sentence, label pair be $X, Y$.

**Assumption 1.** *We assume that from $X$, we can extract two features $X_1$ and $X_2$.*

**Assumption 2.** *$X_1$ is a spurious feature that has strong association with $Y$. Specifically, it means that, solely relying on $X_1$, one can predict with high accuracy over the data distribution, but cannot be 100% correctly.*

**Assumption 3.** *$X_2$ is a robust feature based on which $Y$ can be predicted with 100% accuracy. Namely, there exists a deterministic mapping $f_{X_2 \to Y}$ that maps $X_2$ to $Y$.*

The assumptions above are realistic in some NLP tasks. In NLP tasks, the input $X$ is a sequence of tokens. Some tasks satisfy Assumption 1: $X$ can be decomposed into $X_1$ and $X_2$, where $X_1$ is the presence of certain tokens, and $X_2$ is the context of the token. Thus, $X_2$ has a much higher dimensionality than $X_1$. As shown by the analysis of Gardner et al. (2021), there are indeed datasets where Assumption 2 is true. However, if Assumption 3 is true, we would desire the model to rely on $X_2$, which contains the semantics of the input $X$.

With these assumptions, in Section 2 we empirically demonstrate that spurious features in discrete inputs can cause problems as in the continuous cases Shah et al. (2020); Kalimeris et al. (2019). We show that, possibly due to the simplicity bias, a deep model is likely to excessively rely on $X_1$ and to rely on $X_2$ less. In Section 3.1 and Section 3.2 we provide a theoretical explanation of how MLM pretraining makes a model robust to spurious features. Let $\Pi_1$ be the conditional probability $P(X_1|X_2)$. We show (1) the relation between the mutual information $I(\Pi_1; Y) \geq I(X_1; Y)$ and that (2) the convergence rate of learning from $\Pi_1$ is of the same order as learning from $X_1$. That is, when the MLM model can perfectly model the probability $P(X_1|X_2)$ and thus generate perfect $\Pi_1$, learning from $\Pi_1$ is as easy as learning from $X_1$. As a result, the model will be more likely to rely on $\Pi_1$. Since $\Pi_1$ is estimated based on $X_2$, higher reliance on $\Pi_1$ also implies higher reliance on the robust feature $X_2$. This avoids the pitfall of simplicity bias that the model relies excessively on $X_1$. To relax Assumption 3, we make one step further by considering causal settings in Section 3.3.

The above results partly explain why MLM pretrining is useful for NLP. Denote a sequence of tokens as $X = \langle X_1, X_2, \cdots, X_L \rangle$. During the MLM pretraining process, each token is masked randomly at a certain probability, and the training objective is to predict the masked tokens with the maximum likelihood loss. As a result, the model is capable of estimating the conditional probability $P(X_i|X \setminus X_i)$ for all $i = 1, 2, \cdots, L$. Even though which of the tokens is spurious is unknown, as long as the spurious token has a non-zero probability to be masked during pretraining, MLM can estimate its distribution conditioned on the context and thus can reduce the reliance on it.

Finally, we close the gap between our theories and reality. One major gap is that, in reality, we do not use the conditional probability for downstream tasks. Instead, we feed the input $X$ without masking any token and fine-tune the model along with a shallow layer over its output. Regardless of that, we hypothesize that the robustness brought by MLM pretraining still exists. To prove that, in Section 4 we use the toy example and verify the effect of MLM pretraining when using the common practice for fine-tuning. In Section 5, we validate our theories with two real-world NLP tasks.

In sum, our study leads to new research directions. Firstly, we provide a new explanation of MLM pretraing's efficacy. Unlike the previous purely theoretical studies Saunshi et al. (2021); Wei et al. (2021), our assumptions are milder and more realistic. Secondly, we study NLP robustness from the perspective of self-supervised model, which has been widely used since Word2vec Mikolov et al. (2013) and thus is indispensable to the generalization to unseen data. We reveal the mechanism that leads to its robustness, which may enable us to further reinforce it in the future.

## 2 A TOY EXAMPLE

To show that spurious association can cause difficulty of convergence, we construct a toy example with random variables $X_1, X_2, Y$ and experimental variables $(d_2, \nu)$. In our setting, the random variables $X_1, X_2, Y$ satisfy the assumptions mentioned above, and the experimental variables $\nu, d_2$ control the the strength the spurious association and the difficulty of learning from the robust features. Finally, we measure the difficulty of the task for different $(d_2, \nu)$ by counting the number of updates required for a model to converge.

Specially, we design the relationship between the random variables in the following way. Let the dimension of the random variables $X_1$ and $X_2$ be 2 and $d_2$ respectively. Their value $x_1 \in \mathcal{X}_1 = \{e_1, e_2\}$ and $x_2 \in \mathcal{X}_2 = \{e_1, \cdots, e_{d_2}\}$, where $e_i$ is the one-hot vector whose $i$th element is 1. We

| $d_2$ | $\nu$ | 1 layer w/o | 2 layers w/o pre | 2 layers w/ pre | 3 layers w/o pre | 3 layers w/ pre |
|---|---|---|---|---|---|---|
| 50 | 0.04 | 3680 (189.5) | 691 (55.8) | 614 (169.1) | 302 (47.2) | 249 (53.7) |
| | 0.10 | 2664 (121.2) | 530 (30.6) | 441 (134.9) | 242 (27.6) | 180 (37.5) |
| | 0.25 | 1420 (96.0) | 352 (23.8) | 300 (62.0) | 179 (13.8) | 148 (28.7) |
| | 0.50 | 306 (79.8) | 141 (40.7) | 118 (33.4) | 106 (23.1) | 89 (24.0) |
| 100 | 0.04 | 5466 (170.1) | 945 (57.2) | 689 (225.3) | 431 (51.1) | 275 (72.1) |
| | 0.10 | 3789 (99.2) | 677 (32.2) | 478 (142.9) | 317 (30.3) | 208 (44.3) |
| | 0.25 | 1952 (64.9) | 428 (13.1) | 330 (85.0) | 214 (16.2) | 169 (32.5) |
| | 0.50 | 330 (78.0) | 156 (34.0) | 133 (41.2) | 128 (28.2) | 112 (36.1) |
| 500 | 0.04 | 11127 (265.9) | 1953 (112.5) | 857 (442.6) | 792 (69.8) | 431 (88.4) |
| | 0.10 | 7912 (169.2) | 1279 (67.5) | 657 (234.9) | 550 (46.7) | 402 (97.0) |
| | 0.25 | 4321 (152.3) | 772 (35.5) | 501 (133.5) | 399 (42.3) | 391 (66.0) |
| | 0.50 | 576 (150.0) | 392 (70.2) | 407 (81.1) | 367 (69.1) | 386 (80.0) |

Table 1: The number of iterations a model w/ or w/o pretraining requires to converge. The number is the average of 25 runs with different random seeds, and the number in parentheses is the standard deviation.

make $X_1$ depends only on $X_2$, so it is not a causal feature of $Y$. We control the strength of the association between $X_1$ and $Y$ with $\nu < 0.5$, making $X_1 = Y$ with probability $1 - \nu$. Given a $\nu$, we design the relation between $X_1$ and $X_2$ in a way that maximize the number of dimensions in $X_2$ containing no information about $X_1$. Denote with $\dot{X}_2$ the middle $2\nu d_2$ dimensions of $X_2$, i.e. the $\lfloor d_2/2 - \nu d_2 \rfloor$th to the $\lfloor d_2/2 + \nu d_2 \rfloor$th elements in $X_2$, the random process is as follows

$$
\begin{aligned}
X_2 &= e_i, i \sim \text{Uniform}(1, d_2) \\
Y &= \begin{cases} -1 & \text{if } X_2 = e_i \text{ for some } i < d_2/2 \\ +1 & \text{otherwise} \end{cases} \\
X_1 &= \begin{cases} e_i, i \sim \text{Uniform}(1, 2) & \text{If } \dot{x}_2 \neq 0 \\ f(X_2) & \text{Otherwise} \end{cases}
\end{aligned}
\tag{1}
$$

where $f(X_2) = e_1$ if $X_2 = e_i$ for some $i < d_2/2$, and $f(X_2) = e_2$ otherwise [1]. In this way, predicting $Y$ solely based on the spurious feature $X_1$ can achieve accuracy $1 - \nu$.

We then inspect the effect of using different $(\nu, d_2)$, the strength of spurious association between $X_1$ and $Y$ and the difficulty of learning from $X_2$. Since it is a linearly separable problem, we choose linear networks as our model. We draw training samples $([X_1; X_2], Y)$ pairs from the random process defined in 1. We use Adam optimization with learning rate 0.001 and the cross-entropy loss. In addition to single-layer linear networks, we also try over-parameterized 2-layer and 3-layer linear networks. The hidden size is $[10, 32]$. Since the task is linear separable and our model is also linear, we can check whether the learned weight can lead to 100% accuracy in the defined distribution. We check it every 25 iterations. We say a model has converged if it is 100% accurate for 5 consecutive checks. We report the number of the iterations required before it converges for different $\nu$ and $d_2$.

Even though it is a linear-separable convex optimization problem, our results in Table 1 show that the strength of the spurious association can impact the number of iterations required to converge. We observe that when $\nu < 0.5$, the models tend to be trapped by the spurious feature, sticking at accuracy $1 - \nu$ for iterations. When the spurious relation between $X_1$ and $Y$ is stronger, i.e. $\nu$ is smaller, the number of iterations required to converge is larger. In addition, the number of iterations is also larger when the $d_2$ is larger. An intuitive explanation is that the learning signal from $X_2$ is more sparse when $d_2$ is larger.

---

[1] $\text{Uniform}(a, b)$ is the uniform distribution over $\{n\}_{n=a}^{b}$.

## 3 A Theoretical Explanation of the Efficacy of MLM Pretraining

### 3.1 $P(X_1|X_2)$ is More Informative Than $X_2$

The toy example above motivates us to consider the information contained in $P(X_1|X_2)$. In the toy example, when predicting $P(Y = 0|X)$, if we simply output $P(X_1 = e_1|X_2)$, then the accuracy of our prediction of $Y$ will be as high as predicting $Y$ solely based on $X_1$. It motivates us to inspect the reliability of the estimated $P(X_1|X_2)$ as a feature for the prediction of $Y$ compared to $X_1$. Let $\Pi_1$ be a $|\mathcal{X}_1|$-dimensional random variable whose value is $P(X_1|X_2)$ [2]. We can prove that when $P(X_1|X_2)$ is estimated perfectly, $\Pi_1$ is at least as informative as $X_1$.

**Lemma 1.** *When $X_1, X_2$ are discrete, if $\Pi_1$ perfect, namely the value of $\Pi_1$ is exactly $P(\cdot|X_2)$, then the mutual information $I(X_1; \Pi_1) = I(X_1; X_2)$. (Proof: Appendix A.1)*

Compared to previous works Hjelm et al. (2019); Belghazi et al. (2018); Oord et al. (2018); Kong et al. (2020) that show some self-supervised training objectives are lower bounds of the mutual information $I(X_1; X_2)$, we directly show that the output of the MLM, $\Pi$, maximizes the mutual information, since $I(X_1; f(X_2)) \leq I(X_1; X_2)$ for any $f$. Moreover, instead of explaining the efficacy of pretraining with the infomax principle Linsker (1988); Bell & Sejnowski (1995), our theories below provide a different perspective.

**Theorem 1.** *If $\Pi$ is perfect,*

$$I(\Pi; Y) \geq I(X_1; Y) \tag{2}$$

*Proof.* Since $\Pi$ is perfect, by Lemma 1, we have

$$I(X_1; X_2) = I(X_1; \Pi). \tag{3}$$

By data processing inequality, Equation 3 implies $I(X_1; X_2|\Pi) = 0$. By Assumption 3, a deterministic mapping $f_{X_2 \to Y}$ from $X_2$ to $Y$ exists. Applying data processing inequality again, we have

$$I(X_1, X_2|\Pi) \geq I(X_1, f_{X_2 \to Y}(X_2)|\Pi) = I(X_1, Y|\Pi) \geq 0, \tag{4}$$

which implies $I(Y, X_1|\Pi) = 0$. Accordingly,

$$H(Y|\Pi) = H(Y|X_1, \Pi) \leq H(Y|X_1) \tag{5}$$

$\square$

Theorem 1 shows that $\Pi$ is a more informative feature than $X_1$. However, a model does not necessarily rely more on a more informative feature if it is not easy. We will discuss more in the next section.

### 3.2 Learning from $\Pi$ is Easy

It is important that learning from $\Pi$ is easy. Because of simplicity bias, a neural network model is likely to rely on the easy-to-learn features Shah et al. (2020); Kalimeris et al. (2019). We conjecture that a model excessively relies on the spurious feature $X_1$ when learning from $X_1$ is easier than learning from the robust feature $X_2$. If learning from $\Pi$ is easy, then the model will rely on $\Pi$ more and thus will rely on $X_1$ less. However, features with higher mutual information to $Y$ are not necessarily easy to learn. For instance, although $X_2$ is more informative, models tend to rely on $X_1$ instead of $X_2$ at the beginning of the training process. To show that MLM can mitigate the issue brought by simplicity bias, we need to show learning from $\Pi$ is easy.

Therefore, we have the following theorem that implies learning from $\Pi$ is at least as easy as learning from $X_1$:

**Theorem 2.** *Let $\tilde{h}_{X_1}^{(D_n)} : \mathcal{X}_1 \to \mathcal{Y}$ be the classifier trained with MLE loss using $n$ data pairs $(x_1^{(1)}, y^{(1)}), (x_1^{(2)}, y^{(2)}), \cdots (x_1^{(n)}, y^{(n)})$, and the converged classifier be $\tilde{h}_{X_1}^*$. There exists a learning*

---

[2] We will omit the subscript of $\Pi_1$ when there is no ambiguity.

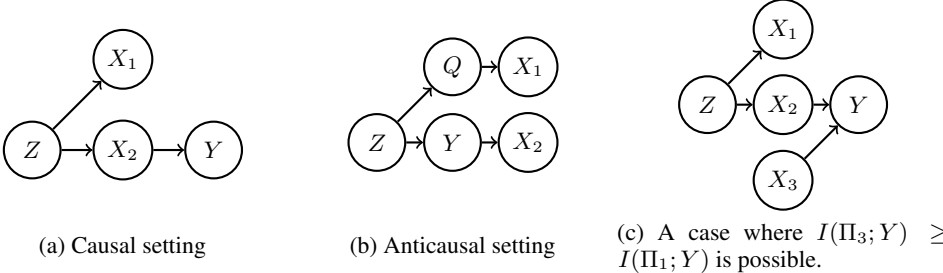

(a) Causal setting      (b) Anticausal setting      (c) A case where $I(\Pi_3; Y) \geq I(\Pi_1; Y)$ is possible.

Figure 2: The causal settings of the $(X, Y)$ pairs.

|  | Ours | Wei et al. (2021) |
|---|---|---|
| structural assumption | $X, Y$ follow Figure 2b. | $X, Y$ follow an HMM. |
| linear independence assum. | $\{P(X_1\|y)\|y \in \mathcal{Y}\}$ | $\{P(X_0\|H_0 = h)\|h \in \mathcal{H}\}$ |
| implication | $I(P(X_1\|X_2); Y) = I(X_2; Y)$ | $I(P(H_0\|X); Y) = I(X; Y)$ |

Table 2: Comparison between Theorem 4 in this work and Theorem 3.3 in Wei et al. (2021).

*algorithm, which generates $\tilde{h}_{\Pi}^{(D_n)} : \Pi \to \mathcal{Y}$ using $(\Pi_1, y^{(1)}), (\pi^{(2)}, y^{(2)}), \cdots, (\pi^{(n)}, y^{(n)})$, such that the following three properties are satisfied: (1)*

$$\mathbb{E}\left[D_{KL}\left[\tilde{h}_{\Pi}^{(D_n)} \,\Big\|\, \tilde{h}_{\Pi}^*\right]\right] = O\left(\frac{1}{n}\right), \tag{6}$$

*which is asymptotically at the same rate as $\mathbb{E}\left[D_{KL}\left[\tilde{h}_{X_1}^{(D_n)} \,\Big\|\, \tilde{h}_{X_1}^*\right]\right]$. (2) Over the distribution of $(X, Y)$, the expected loss of the converged classifier $\tilde{h}_{\Pi}^*$ is not greater than the expected loss of $\tilde{h}_{X_1}^*$. (3) $\tilde{h}_{\Pi}^*$ is a linear model, whose input is $\Pi$. (Proof: Appendix A.2)*

The remaining question is whether deep learning models used in common practices can perform at least as well as the algorithm in Theorem 2. Indeed, without any knowledge of deep learning models, it is impossible to theoretically prove that a model will necessarily rely on $\Pi$ instead of $X_1$. Therefore, in Section 4 and Section 5 we will empirically validate that our theorems are applicable in the real world scenarios.

### 3.3 EXTENDING WITH CAUSAL MODELS

We make a step further by relaxing Assumption 3. By treating $X_1$ as a confounder we can see how MLM pre-training is helpful in the causal and anticausal settings as in Kaushik et al. (2021).

**Theorem 3.** *Even if Assumption 3 is not true, Theorem 1 still holds if $X_1, X_2, Y$ follow the causal setting in Figure 2a.*

*Proof.* By the structure of $X_1, X_2, Y$, inequality 4 holds even if the deterministic mapping $f_{X_2 \to Y}$ does not exist. $\qquad\square$

**Theorem 4.** *Assume that the set of vectors $\{P(X_1|Y = y)|y \in \mathcal{Y}\}$ is linear independent, and if $X_1, X_2, Y$ follow the anticausal setting in Figure 2b, then $I(\Pi; Y) \geq I(X_2; Y)$.*

*Proof.* The assumption is a special case of the one in Lee et al. (2020), so similar techniques can be used: According to the structure of $X_1, X_2, Y$, we have

$$P(X_1|X_2) = \sum_y P(X_1|y)P(y|X_2). \tag{7}$$

Therefore, if $\{P(X_1|Y = y)|y \in \mathcal{Y}\}$ is linearly independent, $P(y|X_2)$ can be recovered from $\Pi = P(X_1|X_2)$. $\qquad\square$

Note that this theorem is very similar to Theorem 3.3 in Wei et al. (2021). However, the assumptions required in ours are weaker and more realistic, and the implication is very similar (Table 2): **(1) Structure assumption:** Wei et al. (2021) assumed that $X$ is generated from a HMM process with hidden variables $H_0, H_1, \cdots$, which is stronger assumption than our assumption that $X_1, X_2$ follow the anticausal setting. **(2) Independence assumption:** Wei et al. (2021) assumed that the vectors in $\{P(X_0|H_0 = h)|h \in \mathcal{H}\}$ need to be linearly independent. In comparison, we require only the independence in $\{P(X_1|Y = y)|y \in \mathcal{Y}\}$. Our assumption is more realistic because the number of hidden states $|\mathcal{H}|$ must be very large if $X$ is generated from the HMM model, and $|\mathcal{Y}|$ tends to be much smaller than $|\mathcal{H}|$. For example, in binary classification cases, our assumption holds as long as $P(X_1)$ is not independent of $P(Y)$. **(3) Implication:** If we further assume that $I(X_2; Y) = I(X; Y)$, then we reach a similar conclusion that $P(Y|X)$ can be recovered from $\Pi = P(X_1|X_2)$ by applying a linear function.

### 3.4 Limitations of Our Theorems

Our theories do not ensure that $\Pi_1$ is the most informative feature to learn from. Consider tokens in a sentence $X = \langle X_1, X_2, \cdots, X_L \rangle$ and let $\Pi_i$ be the conditional probability $P(X_i|X \setminus X_i)$. A token with spurious association with the label can locate arbitrary position in the sentence, and its location is unknown during pretraining. That is, the pretrained model is able to generate $\Pi_i$ for all $i$. Without loss of generality, assume $X_1$ is the spurious token. *It is possible that there exists some $i$ such that $I(\Pi_1; Y) < I(\Pi_i; Y)$, and that $\Pi_i$ is predicted relying on $X_1$.* Concretely, here is an example for the causal setting with three features: $X_3$ is independent of $X_1$ and $X_2$ given $Y$ (Figure 2c). Using the results in Theorem 4, there is a linear mapping that can recover $P(Y|X_1, X_2)$ from $\Pi_3$. Therefore, it is possible that $I(\Pi_3; Y) > I(\Pi_1; Y)$ if $I(X_1, X_2; Y) > I(\Pi_1; Y)$ depending on the distribution of the data. We leave the study of $I(\Pi_i; Y)$ for future work.

Another limitation is that, in practice, NLP practitioners do not use the conditional probability predicted by the pretrained model. Instead, people stack a simple layer over the pretrained model, and fine-tune the whole model on downstream tasks. Regardless of this, we conjecture that the representation encoded by an MLM pretrained model still contains the information of $\{\Pi_i\}_{i=1}^{n}$ and thus is robust to spurious lexicon-level feature.

## 4 Toy Example with a Pretrained Model

As the first step to close the gap between our theories and the real world, we repeat the toy experiments with pretraining. Before fitting the model with $Y$, we first pretrain the first layer to predict $X_1$ based on masked $X$. What we want to show is that, after pretraining, the representation encoded by the layer will have the equivalent role of $\Pi$ even when the input is not masked.

Specifically, the experimental design is as follows: We use the two-layer and three-layer MLP architectures same as in Section 2. When pretraining, we mask $X_1$ in $X$ by using $X' = [0, 0; X_2]$ as inputs. Let the output from the first linear layer as $Z = WX'$. The loss function is the cross-entropy between $X_1$ and the softmax over $[z_1, z_2]$. After pretraining, we fine-tune the pretrained model with $([X_1; X_2], Y)$ pairs, and report the average number of iterations required to converge for 25 different random seeds.

We want to eliminate the possibility that the faster convergence of the pretrained model is because of larger initial weights over $X_1$. Therefore, after pretraining, we manually create a path from $X_1$ to $Z$. We do so by initializing the weights of the third and fourth row of $W$ with $[k, -k, 0, \cdots, 0]$ and $[-k, k, \cdots, 0, 0]$ respectively, where $k$ is the average of the absolute value of the weights in the pretrained part, i.e. the weights of the first two rows in $W$. In this way, the information from $X_1$ has the same scale as the pretrained representation $[Z_1, Z_2]$, and thus it can compete with $[Z_1, Z_2]$ fairly.

Table 1 shows that pretraining can always reduce the number of iterations required to converge when $\nu < 0.50$. The effect is more significant when $d_2$ is larger. It could be because of the higher sparsity of the learning signal from $X_2$ when $d_2$ is larger.

We further inspect how the importance over the inputs changes in the process of training. The importance can be inferred from the product of the linear layers. We observe that if the model is not pretrained, the weights over $X_1$ grow faster than the weights over $X_2$ at the beginning (the

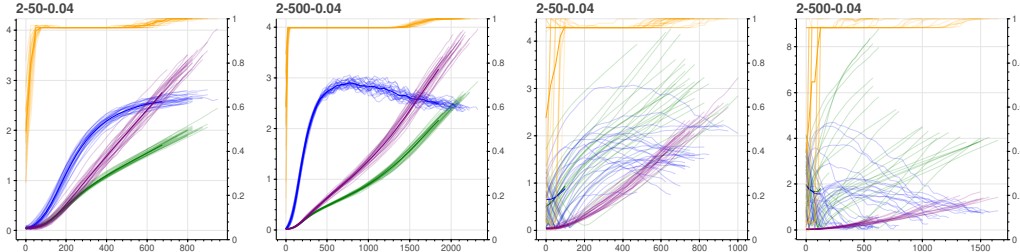

Figure 3: The average weights over the features during training a two-layer model. From left to right, $(d_2, \nu) = (50, 0.04), (500, 0.04), (50, 0.04), (500, 0.04)$. The two figures left/right side are the curves of model without and with pretraining respectively. Blue, green, purple curves represent the average weights over features in $X_1$, $X_2$, and $\dot{X}_2$ (the middle part of $X_2$) respectively. The orange curve represents the accuracy.

first row Figure 3). The model cannot converge to 100% accuracy until weights on $\dot{X}_2$, the middle $\lceil \nu \times d_2 \rceil$ dimensions of $X_2$, become greater than the weights on $X_1$. In addition, after the model converges, weights over $X_1$ is still greater than weights over $X_2$. On the other hand, if the model is pretrained, weights over $X_1$ stop growing after a few steps (the second row in Figure 3). The above observations are aligned with our conjecture that the pretrained representation mitigates the robustness issue brought simplicity bias.

## 5 EXPERIMENTS

We experiment on real world NLP tasks to verify the relation between the capability of modeling the distribution of spurious features $\Pi_1$ and robustness. We facilitate datasets with known spurious features. We first pretrain models on the training dataset with different masking policies. One of them does not mask the spurious tokens, leading to the reduced capability of modeling $\Pi_1$. Afterward, we fine-tune the models using the target label and evaluate their robustness on a testing set where the spurious association does not exist. Our results show that the models are less robust if spurious tokens are not masked during pretraining. This validates our theories that the capability of modeling the distribution of spurious features is important for the models' robustness.

### 5.1 DOWSTREAM TASKS

**Hate Speech Detection**   Previous study has shown that hate speech detection datasets tend to have lexical bias Dixon et al. (2018). That is, models rely excessively on the presence or the absence of certain words when predicting the label. Here we follow the formulation of lexical bias in hate speech detection proposed by Zhou et al. (2021). We focus on the effect of non-offensive minority identity (NOI) mentions, such as "woman", "gay", "black". Those mentions are often highly associated with hateful instances. However, it is more desirable that a model does not rely on those mentions. Therefore, we can see the presence of NOI as a spurious feature.

**Name Entity Recognition (NER)**   Lin et al. (2020) has shown that name entity recognition (NER) models perform worse when the name entities are not seen in the training data. In this case, we can see the content of the name entities as a spurious feature. Models may learn to memorize the name entities when fitting the training data, while we may desire the model to recognize name entities according to the context.

### 5.1.1 DATASETS

**Hate Speech Detection**   We use a portion of the dataset proposed by Founta et al. (2018). In their original dataset, only a small number of hateful instances contain NOI. Our preliminary experiments show that the model without pretraining does not suffer much from the bias of NOI when training with the full data. Therefore, we create a dataset, whose positive (hateful) instances are all the positive samples in the original dataset that contain NOI. As for negative instances, we sample them randomly

| Mask Policy | NER | | Hate Speech Detection | | | |
| | Origin | Unseen | All (12893) | | NOI (602) | |
| | F1 ↑ | F1 ↑ | Accuracy ↑ | F1 ↑ | Accuracy ↑ | FPR ↓ |
| --- | --- | --- | --- | --- | --- | --- |
| scratch | 61.5 $_{0.5}$ | 38.7 $_{0.6}$ | 83.9 $_{1.6}$ | 80.3 $_{1.4}$ | 74.8 $_{1.5}$ | 46.3 $_{7.2}$ |
| vanilla | 74.2 $_{0.4}$ | 56.5 $_{1.3}$ | 83.1 $_{0.8}$ | 78.5 $_{0.8}$ | 75.8 $_{0.5}$ | 25.1 $_{1.8}$ |
| unmask random | 72.7 $_{0.6}$ | 56.5 $_{0.8}$ | 83.3 $_{1.1}$ | 78.9 $_{1.1}$ | 75.8 $_{0.9}$ | 25.7 $_{2.3}$ |
| unmask spurious | 72.9 $_{0.5}$ | 53.2 $_{0.8}$ | 84.1 $_{0.7}$ | 79.8 $_{0.6}$ | 73.7 $_{1.0}$ | 32.5 $_{2.1}$ |
| remove spurious | 69.8 $_{0.5}$ | 56.7 $_{0.8}$ | 82.4 $_{1.0}$ | 77.8 $_{1.0}$ | 77.3 $_{0.6}$ | 21.7 $_{2.0}$ |

Table 3: The performance on downstream tasks. For the hate speech detection task, we also report false positive detection (FPR) on the NOI subset, which is a set of instances containing non-offensive minority identity mentions, e.g. "women", "black". The results are the average of 5 runs, and the smaller number is the standard deviation.

from the original training set. We control the number of negative instances so the ratio of positive and negative instances is the same as the original dataset. We create both the training and the validation splits in this way, and use the original full testing set for evaluation. We also evaluate the models on a NOI subset where all the instances contain NOI. Note that the testing set is from the real world, so the NOIs in the testing set do not spuriously associate with label as strong as in the training set.

**NER** We use the standard NER dataset Conll-2003 Tjong Kim Sang & De Meulder (2003). To create a testing set with name entities unseen in the training set, we replace the name entities in the original validation and testing splits with the entities from WNUT-17 Derczynski et al. (2017). Specifically, we replace the `LOC`, `ORG`, `PER` entities with the corresponding type of entities in WNUT-17, while the `MISC` entities remain untouched.

## 5.2 MASKING POLICIES

For each sentence with $n_s$ spurious tokens, we experiment with different masking policies: **(1) scratch**: We do not pretrain the model before fine-tuning. **(2) vanilla**: During pretraining, we mask each token with 15% probability, which is same as the original implementation in Devlin et al. (2019). **(3) unmask random**: This is similar to vanilla MLM, but we uniformly randomly select $n_s$ tokens from the whole sentence and unmask them if they have been masked. **(4) unmask spurious**: This is similar to vanilla MLM, but we unmask all the spurious tokens. **(5) remove spurious**: We replace spurious tokens with a special "[unk]" token, and we unmask them. Note that this setting can be seen as an oracle setting, since in most applications the spurious features are unknown.

We will inspect the effect of masking spurious tokens by comparing setting (3), (4), (5). Note that these three setting have the same expected number of masked tokens. Therefore, it rules out the possibility that their downstream performance differs because of the number of masked tokens.

## 5.3 IMPLEMENTATION DETAILS

For both of the tasks and all the MLM settings, including the *scratch setting*, we tokenize the input with the bert-base-uncased tokenizer. We use the bert-base-uncased architecture and also the pretrained embedding layer, which is frozen through the pretraining process. We use the existing token embedding to deal with the unseen tokens in the testing set, and this setting is commonly used before contexualized word embeddings were invented. We repeat each experiment 5 times. We include more details in Appendix A.4.

## 5.4 RESULT AND DISCUSSION

Results in Table 3 validate our theorems. For both of the tasks, *unmask random* performs better than *unmask spurious* under distribution shift. Specifically, *unmask random* has higher F1 on the unseen set of the NER task, and *unmask random* has a lower false positive rate (FPR) on the NOI set. Also, *unmask random* performs similarly to *vanilla*. This implies that modeling the condition distribution of spurious tokens in the original random masking pretraining can reduce models' reliance on them.

Note that *unmask random* and *unmask spurious* have similar in-distribution performance, so the performance difference is not due to better in-distribution generalization suggested by Miller et al. (2021).

We also compare *unmask random* with the oracle setting *remove spurious*. We notice that even though *remove spurious* performs as well as *random*, *remove spurious* hurts the performance in the seen set. It indicates that modeling the conditional distribution of spurious tokens has effects beyond simply removing them from the model. On the other hand, *remove spurious* performs better in the hate speech detection task. A possible explanation is that NOI mentions contain little useful information for the task.

# 6 RELATED WORK

Recently, there are efforts attempting to explain the effectiveness of massive language modeling pretraining. Theoretically, Saunshi et al. (2021) explore why auto-regressive language models help solve downstream tasks. However, their explanation is based on the assumption that the downstream tasks are *natural tasks*, i.e. tasks that can be reformulated as sentence completion tasks. Their explanation also requires the pretrained language model to perform well for any sentence completion tasks, which is not likely to be true in the real world. Wei et al. (2021) analyze the effect of fine-tuning a pretrained MLM model. Nonetheless, they have stronger assumptions as described in Section A.4. Aghajanyan et al. (2020) show that pretrained models have lower intrinsic dimension, providing a generalization bound based on Arora et al. (2018). However, why pretrained models have lower intrinsic dimension is unknown. Merrill et al. (2021) show that the parameter norm growth during training makes transformer a saturated model, which can be described in terms of formal languages. Empirically, Zhang & Hashimoto (2021) show that the effectiveness of MLM pretraining cannot be explained by formulating the downstream tasks as sentence completion problems. Sinha et al. (2021) find evidence supporting the hypothesis that masked language models benefit from modeling high-order word co-occurrence instead of word order. There are also some theories explaining the efficacy of non-MLM pretraining Lee et al. (2020); Saunshi et al. (2019); Zhang & Stratos (2021).

Many of the previous studies on robust NLP focus on supervised learning Wang et al. (2021); Utama et al. (2020b;a); Karimi Mahabadi et al. (2020); Chang et al. (2020); He et al. (2019); Sagawa* et al. (2020); Kennedy et al. (2020); Chiang et al. (2020). However, without self-supervised learning, a model can impossibly extrapolate to out-of-distribution data when the domain shifts. Our work also complements previous studies that focus on the bias or robustness of a model generated by the pretraining process Kumar et al. (2020); Hawkins et al. (2020); Vargas & Cotterell (2020); Liu et al. (2020); Gonen & Goldberg (2019); Kurita et al. (2019); Zhao et al. (2019). In this work we investigate the pretraining process itself.

# 7 IMPLICATION AND CONCLUSION

Our results provide possible explanations for some common practices found effective empirically. First, it could explain why continuing pretraining on target dataset is useful Gururangan et al. (2020). It may be because continuing pretrained models model the distribution of spurious features in the target dataset better. Thus the model can better avoid the simplicity pitfall. Second, it provides reasons for more complex masking policies, such as masking continuous random spans Clark et al. (2020); Joshi et al. (2020). It may improve the robustness to spurious features that contain more than one token. Third, if MLM can alleviate the simplicity bias and help the model to achieve a greater margin, it may also imply that the model has wider optima, explaining the finding in Hao et al. (2019). On the other hand, a pretrained model's capability of utilizing more features may also suggest that pretraining makes the decision boundary simpler.

In sum, we show a benefit of MLM pretraining, which partly explains its efficacy. We first empirically demonstrate the presence of simplicity bias when the input is discrete. We then theoretically and empirically explain how MLM pretraining can alleviate the problem brought by it. Finally, we close the gap between our theories and real-world practices with experiments on real-world NLP tasks. Our theories reveal a desirable mechanism of MLM pretraining, suggesting that reinforcing this mechanism could be a promising future research direction.

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
