# OpenReview forum: "On a Benefit of Masked Language Model Pretraining: Robustness to Simplicity Bias"
_ICLR.cc/2023/Conference — Submitted to ICLR 2023_

### Official Review · Reviewer_vBVm · 2022-10-23

**Confidence:** 3
**Correctness:** 2
**Technical Novelty And Significance:** 1
**Empirical Novelty And Significance:** 1
**Recommendation:** 3

**Clarity, Quality, Novelty And Reproducibility:**

* While each small part of the paper (e.g. a subsection describing one theorem, or a subsection describing one experiment) is not difficult to follow, the connection between different parts (e.g. how different theorems in the paper are connected to each other, how empirical experiments are connected to the theories) is not very tied and even not explicitly specified, thus making it difficult to understand the logical flow of the paper.
* Reproducing empirical experiments will be very difficult, even with the information provided in Appendix: (1) on which data MLM pretraining is done is not provided, and (2) details of hyperparameters are not provided (the Appendix only mentions that it follows Huggingface’s implementation, but given that there are many implementations of fine-tuning in Huggingface’s library, this is not very informative. In fact, it is significantly lack of critical information that is necessary to evaluate the validity of the experiments, such as MLM pretraining data, the number of pretraining steps, the final masking ratio (which is a function of n_s in Section 5.2, whose value is not provided), and details in fine-tuning (in fact, whether fine-tuning is done or not is not explicitly mentioned. The paper never mentions fine-tuning, but I don’t think it is possible to achieve reasonable accuracy without fine-tuning (where Y is not given)).


**Strength And Weaknesses:**

### Strengths
The paper addresses an important research question on what makes MLMs useful, which I think is a critical research question, given that MLMs have been widely used in a range of NLP problems.

### Weakness

#### Theory part:
* The summary of the theory part of the paper is “learning from P(X1|X2) is as easy as learning from X1, and is more informative than learning from X2.” The paper then derives the conclusion that MLM training is effective, because it learns from P(X1|X2). I am not convinced with the logical flow here – I don’t think we can say MLM is learning from P(X1|X2) only because in some cases tokens that are masked out belong to X1 and tokens that remain belong to X2.
* In fact, I am not convinced if we can make a distinction between X1 and X2 during MLM training. In MLM pre-training, there is no Y (while one can consider Y as tokens that are replaced with <mask>, this paper is treating all tokens in the input as either X1 and X2). So the distinction between X1 and X2 cannot be made (because their definition depends on Y).
* Moreover, it is not clear to me that in this analogy, what the baseline to MLM is, e.g., MLM is more effective than which method? Based on my best attempt, it looks like the paper is comparing learning from P(X1|X2) vs. learning directly from X1 and X2. But in reality, MLMs are trained with MLM objective and then fine-tuned on the downstream data that directly learns from X1 and X2. So, I am not sure if this analogy makes sense.
#### Empirical part:
* First of all, because the paper provides almost no detail in training (e.g. pretraining data, the number of pretraining steps, details on fine-tuning), it is very difficult to evaluate the validity of the experiments. However, based on my best attempt – it looks like the MLM pretraining is done on the downstream dataset (e.g. hate speech detection task and NER data) rather than unlabeled text corpus, and it is fundamentally different from how pre-training is typically done. In fact, in typical pretraining, there cannot be a notation of spurious tokens / non-spurious tokens since Y is not pre-defined (related to the point I made earlier.
* The conclusion that can be made from the experiments is: “masking tokens randomly at random is better than masking tokens that are not the spurious tokens”. It is not clear to me how this is related to the theory provided earlier in the paper or the overall research question on what makes MLM better. In fact, this result is pretty much expected. The tokens that the paper defines as spurious features (and thus never masked out) are those that are critical in downstream tasks, even if it’s not possible to achieve 100% only with these features. For instance, in NER, the paper defines “entity names” as spurious features, and never masks out entity names. Thus, it is very natural that this pretraining will lead to performance drop in NER, and these results itself do not really verify that predicting spurious tokens from non-spurious tokens is the key to the effectiveness of MLM.


**Summary Of The Paper:**

This paper aims to understand “what makes masked language models (MLMs) useful”. The paper claims that it is because MLM pretraining is specifically effective in reducing the reliance on spurious features, and provides the theoretical view of it. Specifically, it first (1) assumes two random variables from the input, X1 and X2, where X1 is a spurious correlation and X2 is not, and the output Y, (2) shows that the mutual information between P(X1|X2) and Y is larger than the mutual information between X1 and Y, (3) shows that learning from P(X1|X2) is as easy as learning from X1, and (4) claims that this indicates MLM pretraining is effective because it essentially learns from P(X1|X2), since there are instances where non-spurious tokens are masked out, so that the model should predict non-spurious tokens (X2) given the spurious tokens (X1). The paper also empirically shows that, MLM training where spurious tokens are never masked out always give worse results than masking at uniform with the same masking budget, masking a connection between its theory and experiments with real NLP datasets.


**Summary Of The Review:**

Overall, due to the reasons I wrote above as weaknesses, I think the paper needs more work on making a tighter connection between theoretical concepts and the actual MLM training, as well as the empirical experiments with real NLP datasets.

---

> ### Author Response · Authors · 2022-11-17
> **Response to Reviewer vBVm**
>
> Thank you for your reviews. We clarify a few confusions below:
>
> Theory part:
>
> - Our theory is NOT based on the assumption that the pretraining process only mask out X1. Our theory does not require the knowledge about which features belong to X1 or X2 during pretraining either.
> - We admit it is our limitation that in practice we don’t learn from P(X1|X2) directly. But we assume that after pretraining, the model will have this information in its intermediate representation.
>
> Empirical part:
>
> - The experimental design is for the purpose of validating our theorem. Especially, as stated in the paper, we chose those tasks because spurious features are known in those tasks, which allows us to inspect the effect of masking those tokens.
>
> > In fact, this result is pretty much expected. The tokens that the paper defines as spurious features (and thus never masked out) are those that are critical in downstream tasks, even if it’s not possible to achieve 100% only with these features.
>
> - The idea of our experiments is to see how different masking policy will affect the model’s reliance on those features. As shown in the results, when the model does not model the conditional probability for the spurious tokens, the performance is lower. Your argument seems to be based on the assumption that masking an important token is important for the model to learn to utilize the token. However, we could not see why this assumption is obvious.

---

### Official Review · Reviewer_QRgo · 2022-10-25

**Confidence:** 2
**Correctness:** 3
**Technical Novelty And Significance:** 3
**Empirical Novelty And Significance:** 3
**Recommendation:** 5

**Clarity, Quality, Novelty And Reproducibility:**

The paper clearly articulates a proposal for how learning conditional distributions between different features may help overcome simplicity bias, and how the MLM objective can give rise to such learned conditionals. The theoretical setting is different from prior work (for example, Wei et al. make an HMM assumption, while the present paper only assumes that there are features with a causal or anticausal relationship to the classifier label). The paper also introduces and validates the prediction that failing to mask spurious-correlation-inducing tokens during pre-training will undermine robustness, which is a novel and non-obvious observation.

**Strength And Weaknesses:**

Strengths:
- The paper is effective in presenting the high-level idea and intuition behind why having access to conditional probabilities between spurious and robust features can help converge to more robust solutions.
- Experiments in a toy setting provide some support for this claim.
- This explanation gives rise to the falsifiable prediction that pre-training without masking tokens involved in spurious correlations will lead to less robust performance. The paper then conducts experiments targeting this prediction

Weaknesses:
- For the masking experiments in Table 3, it appears that the "unmask spurious" condition for NER involves pre-training where all entity names are defined to be "spurious", and never masked. This seems highly suspect, given how central entity names are to the NER task. That said, other tasks need not exhibit this issue to the same degree. For example in NLI datasets, the presense of the word "not" can be correlated with a "contradiction" label, but simply failing to mask the word "not" doesn't undermine the essense of NLI to the same degree that skipping entity names might undermine NER. Hate speech detection might be a similarly good setting, except that in the "all" condition training from scratch achieves even higher accuracy than nearly all of the pre-training approaches, making it not a suitable testbed. The overall conception of these experiments makes sense to me, but I find the specific task/dataset selection to be inadequate for demonstrating the intended conclusion.
- How do the claims in the paper relate to the known effectiveness of unidirectional LMs? Methods like ELMo and GPT might have slightly lower accuracy than bi-directional models at the same model size, but they seem to capture all of the essential characteristics that make pre-training effective. A robustness explanation that applies to MLM but not to unidirectional LMs would seem incomplete and potentially even missing the mark on what actually matters, and the paper would be stronger if it gave consideration to this issue.
- Some of the claims in the toy setting are perhaps not a strong as the text might suggest. Looking at Figure 3-right, I see many curves where the blue line rises rapidly and does not necessarily exceed the green line after just a few steps. Maybe it's just hard to fully understand the situation with how multiple runs are superimposed on the same plot, but it seems that despite the greater variance given pre-training the relations between the different curves do not necessarily change. For 2-50-0.04, accuracy appears to hit the ceiling at ~600 steps regardless of whether pre-training is used. Results in Table 1 also show that pre-training can sizably increase variance, and increased variance alone could explain any other effects. (cf. how dropout and other methods of adding noise can affect training dynamics)


**Summary Of The Paper:**

The paper investigates the mechanism by which MLM pre-training may result if fine-tuned models that are more robust to spurious features.

The paper proposes a setting where there is a spurious feature that can predict a classification decision with a simple decision boundary and high but non-perfect accuracy, and a robust feature that that can predict the decision perfectly but with a more complex decision boundary. In this setting, the training dynamics of a model are such that it learns to use the spurious feature after a small number of training iterations, but requires many more iterations to make use of the robust feature. Given access to a predicted conditional distribution p(spurious feature | robust feature), however, the effect of the spurious feature is mitigated.

**Summary Of The Review:**

Overall the paper presents a clearly-described theory for how pre-training to learn conditional distributions between features can help prevent over-reliance on spurious features. This proposal is backed up by theoretical claims, experiments on a toy task, as well as by testing one of the theory's predictions in a real NLP setting with pre-training and fine-tuning. However, beyond proving that this theory *can be* an explanation for why pre-training helps, the body of evidence is not large enough to show that this *is in fact* the dominant factor underlying model robustness.

---

> ### Author Response · Authors · 2022-11-17
> **Response to Reviewer QRgo**
>
> Thank you for your review and suggestions!
>
> - For the natural setting of the NER task, we agree that considering named entities as spurious features is not realistic. However, for the purpose of verifying our theory, here we focus more on the OOD setting, where all the named entities are unseen. In this case, though the surface form of those name entities may provide some cue, we believe that they won’t be as informative as in the natural setting. We may consider finding some other tasks for the experiments.
> - We agree that’s a good research direction to think about the essential characteristics that can explain both bidirectional and unidirectional LM. This work can somewhat explain the effectiveness of unidirectional LM though. Concretely, if the robust feature is in the prefix of the spurious one, then modeling P(x_T | {x}_{1}^{T - 1}}) will improve the robustness based on our theory.
> - Thank you for pointing this out. Plotting the difference between the blue lines and the green lines directly would make our claim clearer. Indeed it’s hard to fully understand the dynamic of training, but from Figure 3 we can see that the pretrained model has a higher chance to converge faster. Also, we can see all the pretrained models converge before 1500 steps, while the models without pretraining just start to converge at the 1500th step.

---

### Official Review · Reviewer_RC11 · 2022-11-02

**Confidence:** 4
**Correctness:** 2
**Technical Novelty And Significance:** 3
**Empirical Novelty And Significance:** 3
**Recommendation:** 3

**Clarity, Quality, Novelty And Reproducibility:**

I believe the paper has serious flaws that should be improved before it can be submitted.
However, it is reasonably clear and novel.

**Strength And Weaknesses:**

Strengths:
- A better understanding of MLM pretraining is desirable overall, so the paper's goal is a relevant one.
- Table 1 is partially interesting.

Regarding weaknesses, it seems to me that the paper doesn't make much sense overall, mainly because the overall setting seems too simplified (i.e., neural models don't necessarily rely on clearly separable features). In addition, as the authors themselves note, "in practice, NLP practitioners do not use the conditional probability predicted by the pretrained model. Instead, people stack a simple layer over the pretrained model, and fine-tune the whole model on downstream tasks." The authors claim that this isn't a problem for their thinking in practice, but I'm not convinced this is true.

In addition, I see the following weaknesses:
- The paper is based on the assumption that it's totally unclear why MLM pretraining works ("In this work, as an initial step toward the answer"). This isn't true. Specifically, besides what's mentioned in the related work, I was missing a discussion of the distributional hypothesis.
- The paper is centered around the idea of "spurious features". However, I believe the authors are greatly misusing the term: "spurious" implies that the features are *not*, in fact, good indicators of the final label. This is true for the empirical experiments that are being conducted, but, in the theoretical part, the authors basically construct an imperfect and easy (as opposed to spurious) feature.
- I don't understand why Table 1 has no "with pretraining" column for 1 layer.
- "NER" = "name**d**" entity recognition"
- The authors should not use anything pretrained in their from-scratch experiments. I don't understand why they chose to do this.


**Summary Of The Paper:**

This paper addresses the problem that we don't yet fully understood why masked language model (MLM) pretraining is effective for downstream NLP tasks. What the authors present as (part of) the answer is that MLM pretraining makes models robust to lexicon-level spurious features. They show both theoretical and empirical studies to back up this claim.

Unfortunately, I don't think the paper as-is is in an acceptable state. Specifically, I don't think one can learn anything practically relevant from the paper, as I will explain more under "Weaknesses" below.

**Summary Of The Review:**

While this paper is about an interesting topic, I don't believe the quality is sufficient for it to be published at a top-tier conference. In fact, in its current state it shouldn't be published at all.

---

> ### Author Response · Authors · 2022-11-17
> **Response to Reviewer RC11**
>
> Thank you for your review and suggestions!
>
> - We will include a discussion about the distributional hypothesis in the related work section.
> - Would it possible for you to elaborate more about how the term “spurious features” is misused?
> - We chose to use the token embedding from a pretrained model because for fair comparisons with other settings.

---

### Decision · Program_Chairs · 2023-01-20

**Decision:**

Reject

**Justification For Why Not Higher Score:**

All the reviewers think the paper is not in a good state to be published, so it is a clear rejection to me.

**Justification For Why Not Lower Score:**

N/A

**Metareview: Summary, Strengths And Weaknesses:**

This paper aims to study why MLMs are helpful for downstream tasks. The paper’s main argument is that MLM pre-training can be more robust to spurious features, defined as token-level, simple features.
The reviewers raised the following concerns regarding the assumptions of the work and its experimental settings, including 1) there is a significant gap between this theoretical work and standard practice in NLP (practitioners always fine-tune all the parameters with a newly-added head); 2) all the entity names are considered as spurious features in NER, which is highly suspect; 3) The experiments are only conducted on a toy task; 4) the experimental details are missing, so it seems difficult to reproduce.

Based on all the points, I can’t recommend acceptance in the current form.